# Finite Element Analysis of Reinforced Concrete Beams Prestressed by Fe-Based Shape Memory Alloy Bars

Yeong-Mo Yeon [1], Wookjin Lee [2],* and Ki-Nam Hong [1],*

1 Department of Civil Engineering, Chungbuk National University, 1 Chungdae-ro, Seowon-gu, Cheongju 28644, Korea; yym235@chungbuk.ac.kr
2 School of Materials Science and Engineering, Pusan National University, 2 Busandaehak-ro 43 Beon-gil, Geumjeong-gu, Busan 49315, Korea
* Correspondence: wookjin.lee@pusan.ac.kr (W.L.); hong@chungbuk.ac.kr (K.-N.H.); Tel.: +82-51-510-2997 (W.L.); +82-43-261-2378 (K.-N.H.)

**Abstract:** Prestressing of concrete structures using Fe-based shape memory alloys has been investigated extensively by experiments in the last decade. However, detailed investigations on the stress produced by the Fe-based shape memory alloys and its influence on concrete damage during deformation of concrete structure has not been investigated yet. In this study, the prestressing effect by Fe-based shape memory alloy bars on bending behavior of reinforced concrete beam was investigated numerically. A finite element simulation model was developed to investigated the bending responses of the beams including nonlinear material properties such as concrete cracking and crushing as well as the plastic deformation of the Fe-based shape memory alloy. The model is able to capture the bending behavior of the beam prestressed with the Fe-based shape memory alloy bars. Based on the numerical and experimental results, the prestressing effect by the shape memory alloy bars was investigated in detail. Although the developed model slightly overestimated the experimentally obtained bending load-deflection curves of the concrete beams, it was shown that the developed model can be used for an optimization study to select the best possible design parameters for prestressing the concrete beam with the Fe-based shape memory alloy bars. A possible reason for the overestimation is the idealized perfect bonding assumption between Fe-SMA and concrete used in the model, while slip at the interface occurred in the experiments.

**Keywords:** reinforced concrete; finite element analysis; shape memory alloys; prestress

## 1. Introduction

Shape memory alloy (SMA) is a special alloy with characteristics (Shape Memory Effect (SME)) that can be recovered to its original shape through an activation process consisting of heating and cooling processes even when plastic deformation occurs in it [1]. In 1951, SME was first discovered in Au-Cd single crystal alloy, and many types of SMA have been developed since then [2–5]. In particular, after SME was discovered in Ni-Ti alloy (Nitinol), which was being developed to secure corrosion resistance in 1963, SMA began to be used in various fields such as aircraft, ships, and satellite antennas [6–9].

Recently, many researchers have conducted various studies to use SMA as a construction material [10–15]. SMA with superelasticity can improve the seismic performance of the structure by damping the vibration applied to the structure [16–20]. In addition, SMA enables new attempts that were difficult to implement with conventional construction materials, such as the ability to apply stress to a structure in a desired direction through SMA's SME [21,22]. In particular, a method to introduce prestress using the recovery stress of SMA can resolve several disadvantages of conventional prestressed concrete. When the deformation of the pre-tensioned SMA is constrained and then activated, the SMA tries to return to its original shape, but because the SMA is constrained, a compressive force called recovery stress occurs inside it. As in this principle, when the pre-tensioned SMA

embedded in concrete is activated by induction heating or electrical resistance heating, SME is generated in the SMA and it tries to shrink to its original state. However, since the deformation of the SMA is constrained by the bonding force between the SMA and concrete, recovery stress is generated inside the SMA, and this recovery stress acts as a compressive force on the concrete like prestress force of conventional prestressed concrete.

Concrete members to which prestress are introduced in this method have great mechanical performance because concrete and the SMA work integrally as in the bonded prestressed concrete. In addition, in the method of introducing prestress using SMA, it is possible to simply introduce prestress to the concrete member only by heating SMA without an anchoring device and jacking device, unlike the conventional prestress method. Moreover, unlike conventional bonded prestressed concrete, concrete members to which prestress is introduced through SMA can recover the initial recovery stress through simple reactivation even if recovery stress is lost due to various causes such as drying shrinkage and creep of concrete, and SMA relaxation [23].

The most widely used SMA in the industrial field is Nitinol. However, Nitinol has a relatively narrow temperature history [24]. Moreover, it is practically impossible to use Nitinol, which contains expensive Ni and Ti in large proportions, as a construction material. On the other hand, Fe-based shape memory alloy (Fe-SMA) has a lower manufacturing cost compared to Nitinol because relatively inexpensive iron is used as the main material. Therefore, studies for introducing a prestress force into a structure using Fe-SMA have been studied by many researchers. Sawaguchi et al. [25] conducted an experimental study to evaluate the flexural performance of mortar specimens reinforced with Fe-SMA bars. They reported that the initial cracking load and ultimate load of mortar specimens reinforced with Fe-SMA bars were larger than those of mortar specimens reinforced with stainless steel bars. They explained that the cause was the introduction of a prestressing force into the specimen due to the activation of Fe-SMA. Choi et al. [26] performed an experimental study to confirm the prestressing effect of the Fe-SMA wires in mortar beams. They conducted a bending test of the mortar beams reinforced with Fe-SMA wire. They reported that the crack load, stiffness and ductility of specimens with the activated Fe-SMA wire were increased compared to those of the specimen with non-activated Fe-SMA wire due to the prestress effect. Montoya-Coronado et al. [27] conducted an experimental study to evaluate the shear performance of reinforced concrete (RC) beams shear strengthened by Fe-SMA strips. They reported that the shear strength of RC beams strengthened by the Fe-SMA strip increased by 65~83% compared to those of non-strengthened specimens. They also reported that the RC beams with activated Fe-SMA had a smaller number of diagonal cracks, and increased shear crack load and stiffness compared to the RC beams with non-activated Fe-SMA. Dolatabadi et al. [24] performed an analytical study to evaluate the bending performance of RC beams strengthened with Fe-SMA rebar by the external bonded (EB) method. They presented a finite element (FE) model of an RC member strengthened by the Fe-SMA rebars by the EB method. They validated the FE model by comparing the results predicted by the FE model with the experimental results performed by Shahverdi et al. [28]. They argued that the proposed FE model accurately predicted the bending behavior of the RC member strengthened with the Fe-SMA rebar by the EB method. Yeon et al. [29] performed an analytical study to predict the flexural behavior of RC members strengthened with Fe-SMA strips by the near-surface mounted (NSM) method. They proposed a two-dimensional FE model using OpenSees, and the validity of the FE model was verified by comparing the results predicted by the FE model with the experimental results performed by Hong et al. [30]. They also performed parameter studies to figure out the effects of concrete compressive strength, steel rebar amount, and Fe-SMA reinforcement amount on RC member flexural behavior. Hong et al. [31] conducted an experimental study to evaluate the bending behavior of concrete beams reinforced with Fe-SMA bars. They considered the amount of Fe-SMA bar and Fe-SMA activation as experimental variables. They reported that activation of the Fe-SMA bar applied an eccentric compressive force to the specimen, causing upward displacement in the specimen, and this upward displacement increased as

the amount of Fe-SMA reinforcement increased. In addition, they reported that the initial cracking loads of the specimens in which the Fe-SMA bars were activated were 48–113% greater than those of the specimens in which the Fe-SMA bars were not activated.

A closer examination of the open literature shows that there is limited work on prestressing by Fe-SMAs of RC beams. However, there have been only a limited number of works in the literature dealing with comparisons between the theoretical prestressing effect predicted by numerical models and the experimental results of the RC beam prestressed by the Fe-SMAs. In order to bridge this knowledge gap, the prestressing effect by Fe-SMAs of RC beams was investigated both numerically and experimentally in this study. The numerical simulations were performed through a three-dimensional FE model. The model was developed to trace the response of the RC beam during bending deformation, including nonlinear material properties such as concrete cracking and crushing, as well as the plastic deformation of the Fe-SMAs. The numerical results were compared with experimental results obtained by the bending of the RC beams prestressed by the Fe-SMAs in different prestressing scenarios.

## 2. Experimental Procedures

In the authors' previous work, 10 specimens were constructed to evaluate the bending behavior of concrete beams reinforced with Fe-SMA bars [31]. Figure 1 shows the photos of the Fe-SMA bars, assembly of the Fe-SMA bars and the stirrups, mold for casting the concrete and the concrete casting process for the experiments. The experimental data obtained from the same specimens were used in this study. As shown in Figure 2, the width, effective depth and height of the specimens were 250 mm, 350 mm and 400 mm, respectively. The total length and net-span of the specimens were 2800 mm and 2600 mm, respectively. As described in [31], the previous experimental work considered Fe-SMA activation and amount of Fe-SMA bar (200 mm$^2$, 300 mm$^2$, 400 mm$^2$, and 500 mm$^2$) as experimental variables.

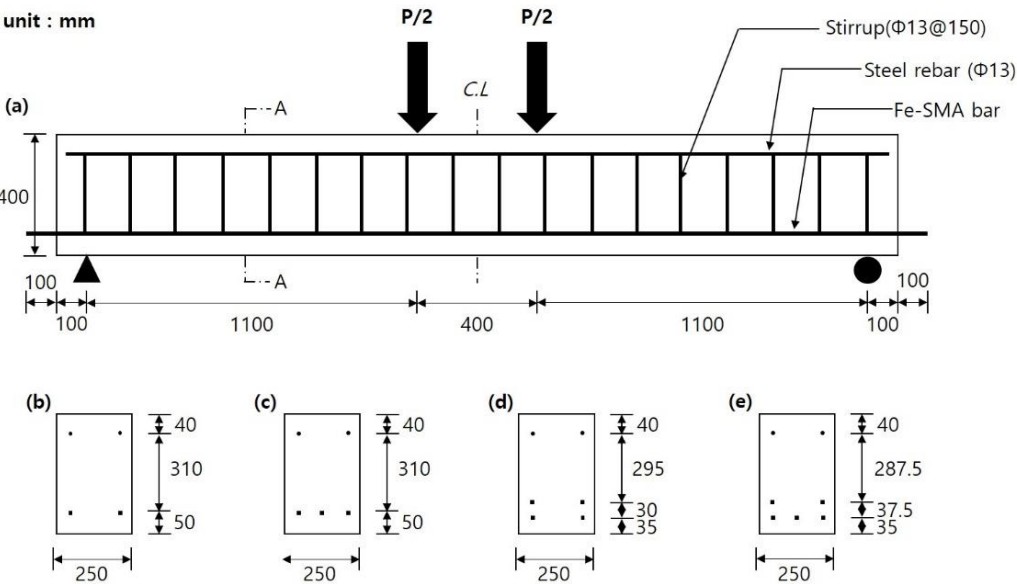

**Figure 1.** Details of test specimens [31]. (**a**) Schematic of test specimens. (**b**) With 2 SMA bars. (**c**) With 3 SMA bars. (**d**) With 4 SMA bars. (**e**) With 5 SMA bar.

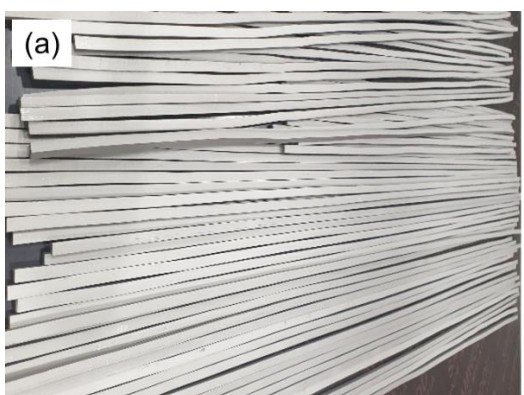 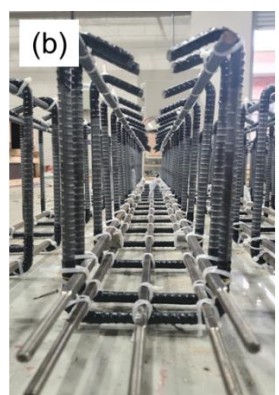 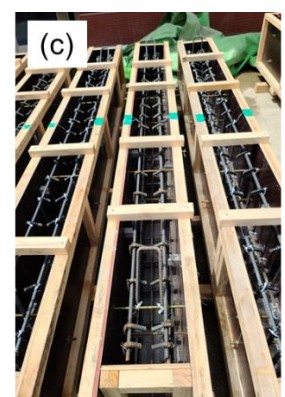 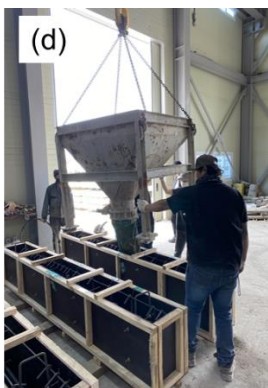

**Figure 2.** Photos of (**a**) Fe-SMA bars used for the experiments, (**b**) Fe-SMA and stirrup assembly, (**c**) molding for concrete casting and (**d**) concrete casting process.

The compressive strength of the concrete used in the specimens was 46.2 MPa. To prevent shear failure, U-shape stirrups with Φ13 mm were placed at a spacing of 150 mm. The compression rebars of the specimens also had the same diameter as stirrups, and the elastic modulus, yield strength, and ultimate strength of the rebar were 200 GPa, 462 MPa, and 540 MPa, respectively. The Fe-SMA bars were stretched to a pre-strain of 0.04 by a horizontal hydraulic jack. The cross-section of the Fe-SMA bar used in the experiment was 10 mm × 10 mm, and the length was 3000 mm. When the Fe-SMA bar was pre-strained to 0.04 and heated to 160 °C, the recovery stress was about 335 MPa.

After the concrete curing was completed, the Fe-SMA was heated to 160 °C by resistance heating. The center displacement of the specimen during Fe-SMA heating and cooling was measured by a linear-variable displacement transducer (LVDT). After the displacement of the specimens became stable, four-point bending tests were carried out to evaluate the flexural behaviors of the specimens.

## 3. Numerical Modeling

The simulations were based on the experimental work by Hong et al. [31]. The commercial FE software package ANSYS [32] was used to develop FE models. Thus, the geometries of the FE models were designed to be the same as the geometries of the specimens used in the previous work [31]. The representative geometry and FE meshes used for the simulations are shown in Figure 3. Half of the geometry of the RC beam reinforced by the Fe-SMAs and rebars was modeled based on the symmetry of the geometry, i.e., the *y-z* plane of the FE model was considered as the plane of symmetry where *x*-direction movements of the FE nodes were constrained during the simulations. The averaged FE mesh size was 25 mm. The bending deformation of the models was simulated by applying *z*-axis displacement on the top of the model where pressure was applied experimentally, while the *z*-axis movement of the supported position was fixed.

Two types of FE elements of SOLID65 and SOLID185 of ANSYS were used in the simulations. The SOLID65 elements [32] used for the concrete material are the 8-noded brick elements with three translational degrees of freedom per node with capability of cracking in tensional load and crushing under compression. The Fe-SMA bars and the rebars were modeled using the 3D structural 8-noded elements of the SOLID185. Perfect bonding between the concrete and the Fe-SMA bars as well as between the concrete and the rebars were assumed throughout the FE simulations.

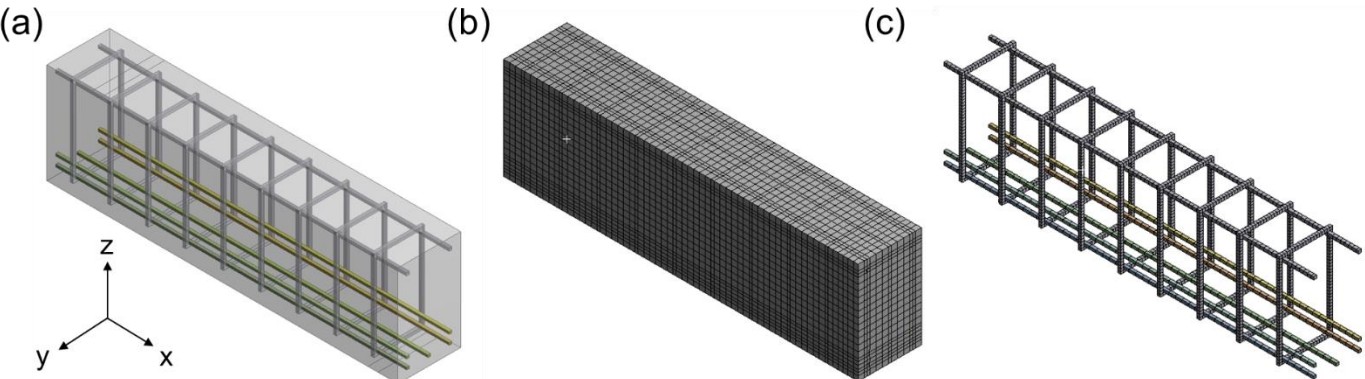

**Figure 3.** Representative geometry and FE meshes of the numerical model, shown for the RC beam model with 5 Fe-SMA bars. (**a**) Geometry of the FE model used in the study for RC beam with 5 SMA bars and FE meshes for (**b**) concrete and (**c**) SMA bars and rebars.

In order to simulate the cracking and non-linear response of the concrete, the constitutive model was based on the theory of William and Warnke [33]. The concrete model used in this study requires five input parameters of uniaxial tensile strength ($f_t$), uniaxial compressive strength ($f'_c$), compressive strength for a state of biaxial compression superimposed on hydrostatic stress state ($f_1$), uniaxial compression superimposed on hydrostatic stress state ($f_2$) and biaxial compressive strength ($f_{cb}$). In this study, the concrete was modeled with $f'_c$ and $f_t$ of 46.2 and 4.28 MPa, respectively, based on the experiments. The other three parameters of $f_{cb}$, $f_1$ and $f_2$ were set based on the original model by William and Warnke [34] as $1.2f'_c$, $1.45f'_c$ and $1.725f'_c$, respectively. Moreover, the values for the open and closed shear crack coefficients of 0.3 and 0.5, respectively, were used for the simulations.

The non-linear compressive stress-strain response of the concrete under uniaxial compression was modeled using the relation proposed by Hognestad et al. [35], as in Equation (1) and shown in Figure 3a.

$$f_c = f'_c \left[ 2 \left( \frac{\varepsilon}{\varepsilon_0} \right) - \left( \frac{\varepsilon}{\varepsilon_0} \right)^2 \right], \tag{1}$$

where $f_c$ is the compressive stress in the concrete at the corresponding strain of $\varepsilon$. $\varepsilon_0$ of Equation (1), which is given as follows:

$$\varepsilon_0 = \frac{2f_c'}{E_c}, \tag{2}$$

where $E_c$ is the elastic modulus of the concrete, which was set to 32 GPa in the FE simulations. The behavior of the concrete in tension was modeled as linear elastic up to the tensile strength of concrete. When the tensile stress of the concrete element reached its tensile strength, the stress relaxation by the cracking was represented by a stepwise drop in the concrete tensile stress by 40%. Then, the curve descended linearly to zero tensile stress at a strain value 6 times higher than the strain where the tensile crack started to appear, similar to a previous work done by Hawileh [35]. For the Fe-SMA bars, an experimentally determined elastic modulus of 150 GPa [31] with Poisson's ratio of 0.33 was used. The non-linear tensile stress-strain response of the Fe-SMA bars was also considered based on the experimentally measured tensile behavior of the material after activating [31], as shown in Figure 3b. The rebars were modeled as the elastic-plastic material with a bilinear isotropic hardening assumption using an elastic modulus of 200 GPa, Poisson's ratio of 0.33, yield strength of 480 MPa and tangent modulus of 20 GPa.

The activation process of the Fe-SMA bars was modeled by applying thermal contractions of the bars along their longitudinal directions. The experimentally obtained recovery

stress under a fully restrained condition for the Fe-SMA bars was 335 MPa [31]. Thus, the thermal contraction of 0.002233 mm/mm was applied for each Fe-SMA bar, which corresponds to the tensile stress of 335 MPa in a fully restrained condition.

## 4. Results and Discussions

### 4.1. Effect of SMA Activation

In this section, the FE model with 5 Fe-SMA bars was used to investigate the effect of Fe-SMA activation on the bending response of the RC beam and the cracking behavior. For this purpose, one FE simulation was conducted without activating the Fe-SMA bars while another simulation was carried out with the activation. The activation of the Fe-SMA bars in this section was done with a single FE step, i.e., all 5 Fe-SMA bars were activated numerically at once. Figure 4 shows the bending load-deflection behavior of the FE models with 5 SMA bars, with and without the activations of the Fe-SMA bars. The bending deformations up to the bending defection of 10 mm was considered to prevent divergency in the FE solutions due to the severe cracking in the concrete.

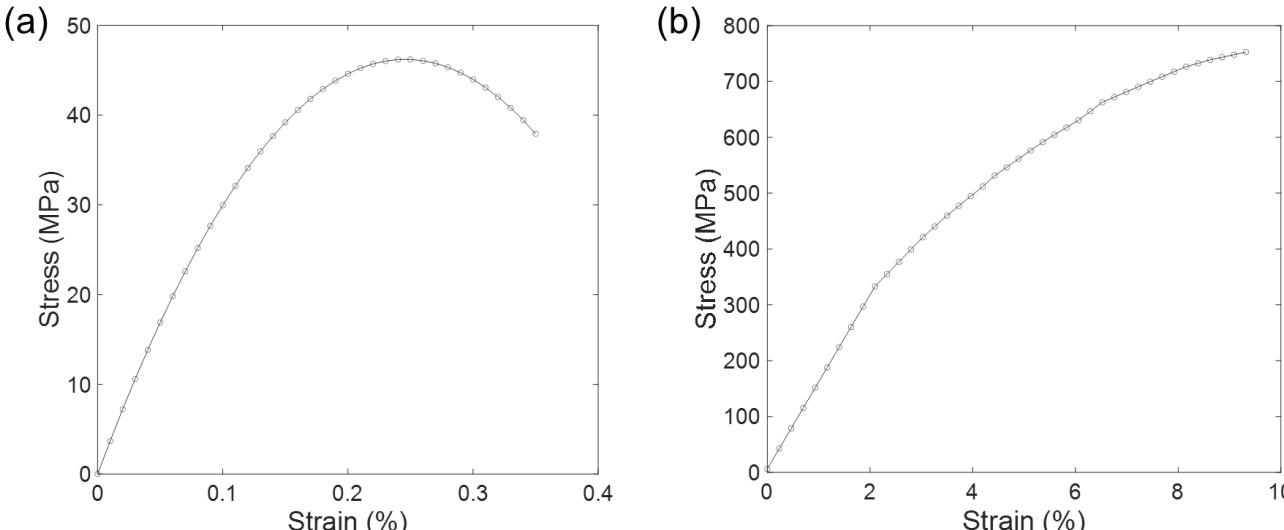

**Figure 4.** Idealized stress-strain curves used for FE simulations. (**a**) Compressive stress-strain curve for concrete and (**b**) tensile stress-strain curve for Fe-SMA bar.

Figure 5 shows the bending load-deflection behavior of the FE models with 5 SMA bars, with and without the activations. In the figure, the experimentally obtained load-deflection curves are also depicted for comparisons. In the FE simulation results, it can be clearly seen that the activation of Fe-SMA bars significantly increased the bending resistance of the RC beam. In the case without Fe-SMA activation, the non-linear response of the simulated load-deflection curve started at the applied load of approximately 50 kN, whereas the non-linearity started to appear at a load of ~130 kN in the case with the Fe-SMA activation. This shows that the developed FE simulation model can effectively take account of the prestressing effect due to the FE-SMA activation.

When comparing the experimentally obtained load-deflection curves with the simulated curves, the FE model without Fe-SMA activation reproduced the experimentally obtained curve very well, as shown in Figure 5. This means that the developed FE simulation model can consider all the major strengthening effects by the Fe-SMA bars and the rebars on the bending deformation of the RC beam when the Fe-SMA bars are not activated. However, the FE model with Fe-SMA activation overestimated the applied load for the bending of the RC beam in comparison to the experimentally obtained behavior by approximately 20–30 kN, in the whole range of the bending deflection tested. Since the only difference between the two FE models (and also between the two RC beams used in the experiments of the bending curves presented in Figure 4) is in the stress states due to

the Fe-SMA bar activations, the reason for the discrepancy between the FE simulation and the experiment in the case of Fe-SMA activation can be due to the inaccurate prediction of the prestressing effect of the Fe-SMA bars.

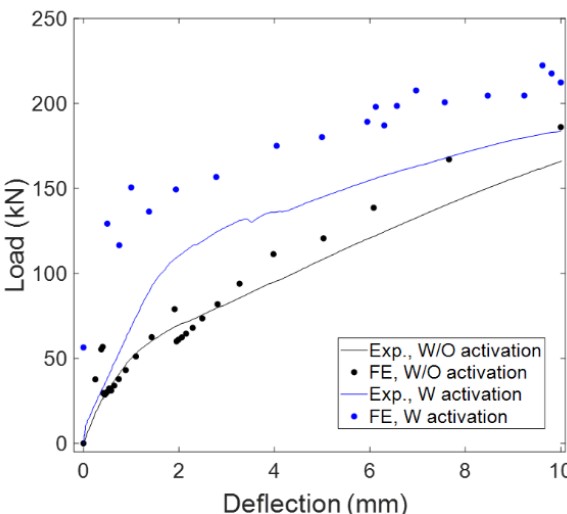

**Figure 5.** Comparison of experimental results and FE simulations for bending load-deflection responses of RC beams with 5 Fe-SMA bars, with (W) and without (W/O) activation of SMA bars.

One possible explanation for the overestimated prestress of the Fe-SMA is the imperfect bonding between the Fe-SMA bars and the concrete in the experiments. In the experiments, there was no anchoring between the concrete and the Fe-SMA bars and the Fe-SMA bars had nearly flat surfaces [31]. If there was debonding or slip of the interfaces between the Fe-SMA bars and the concrete in the experiments, this could make the present FE simulation have an overestimated prestress state while activating the Fe-SMA bars where perfect bonding between the bars and the concrete was assumed. To take into account the realistic bonding behavior between the Fe-SMA bars and the concrete, sophisticated FE simulations with consideration of bonding-debonding [36] or with specialized interface elements for the consideration of slip [35] are needed, which is beyond the scope of current FE simulations.

Although the FE simulation for the RC beam bending with Fe-SMA activation overestimated the experimentally obtained bending load, the FE simulation could capture the overall trend of load-deflection behavior of the experiment, as shown in Figure 4. The FE simulation results predict the onset of non-linear bending deflection at around 1.5 mm deflection due to the cracking of the concrete, which is consistent with the load-deflection behavior observed in the experiment. The slope in the load-deflection curve in the RC beam bending obtained in the FE simulation is in very good accordance to that of the experimentally obtained curve in the wide range of the bending deflection up to 10 mm.

Figure 6 shows the evolutions of cracking and crushing patterns with increasing bending deflection predicted by the FE simulations with and without the activation. From the figure, it can be easily confirmed that the cracks in the RC beam without Fe-SMA activation is more severe and widely spread than those with the activation. At a relatively low bending deflection of 2 mm, the concrete in the RC beam without Fe-SMA activation already showed long and severe cracks mainly along the stirrups near the center of the RC beam. Long vertical cracks along the stirrups as well as inclined shear cracks in between the stirrups were found to be the main cracking mechanisms, as shown in Figure 6. Near the bottom of the RC beam, concrete crushing patterns also appeared, which are due to the localized bending deformation of the cracked concrete elements near the bottom. With increasing the bending deflection, the cracking patterns become more widely spread.

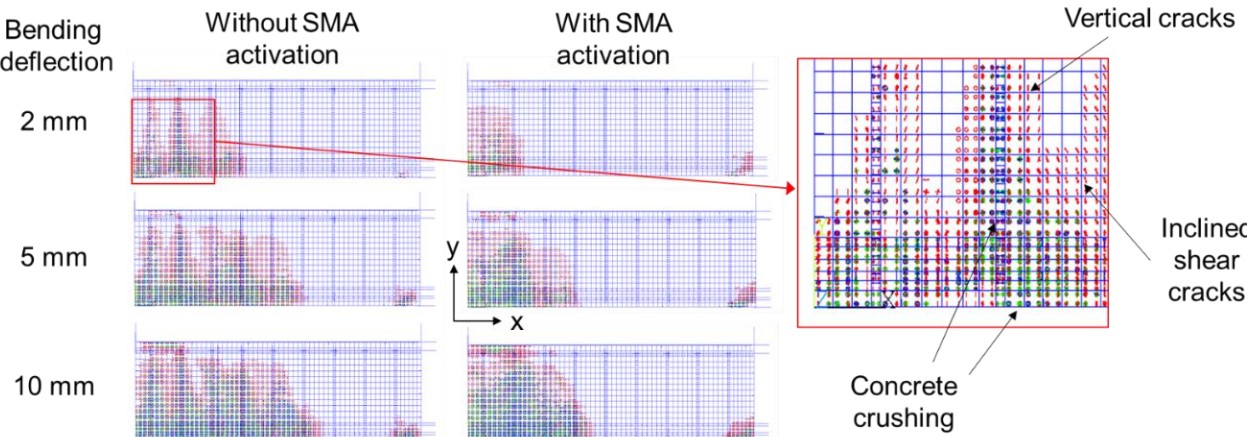

**Figure 6.** Cracking and crushing patterns developed in FE simulation of bending of the RC beam on *x-y* plane, for RC beams with 5 Fe-SMA bars with and without activation.

In comparison to the case without Fe-SMA activation, the RC beam with activated Fe-SMA showed less severe and more localized cracking pattern, as shown in Figure 6. This is due to the prestressing effect by the activation of the Fe-SMA bars, which reduces the tensile stress near the bottom of the RC beam and consequently hinders the onset of cracking in the concrete beam. With increasing the bending deflection, the cracking pattern in the RC beam in this case did not spread far from the center of the RC beam.

Evolutions of equivalent stress fields of the concretes with bending deformations, predicted by the FE simulations of the RC beams with and without Fe-SMA activation are shown in Figure 6. Regardless of the bending deflection, near the top of the RC beam, concrete in the RC beam with the Fe-SMA activation was in a higher stress than that in the beam without the activation. This was because the prestressing associated with the Fe-SMA activation produces additional compressive stress near the top of the RC beam. With increasing bending deflection, the high compressive stress region spreads more widely in the concrete with the Fe-SMA activation than that without the activation, as shown in Figure 7.

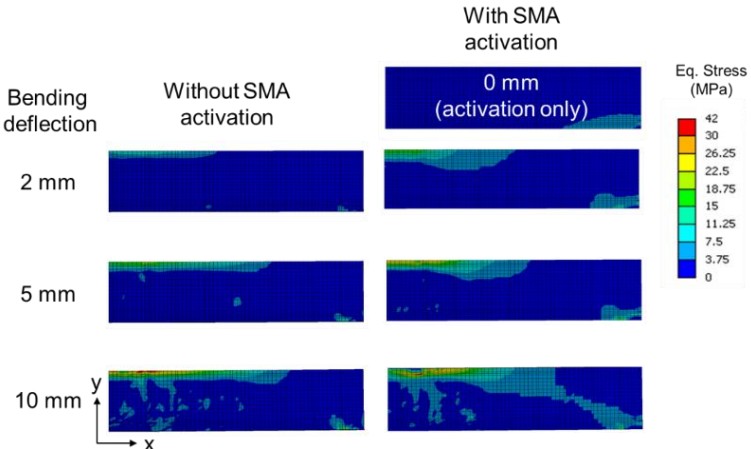

**Figure 7.** Evolutions of equivalent stress fields of concrete with bending predicted by FE simulations of RC beam with 5 Fe-SMA bars with and without activation.

The results shown in Figures 5–7 reflect that the prestressing effect due to the activation of the Fe-SMA bars can be numerically considered with the simple FE model proposed in this study. The FE model with activated Fe-SMA shows much higher resistance to the bending deformation, less severe cracks at the same bending deflection and more widely

spread compressive stress region in the stress fields in the FE solutions than that without Fe-SMA activation.

### 4.2. Effect of Activation Sequence

In the experiments, the Fe-SMA bars were activated by heating them to ~160 °C by electric resistance heating [31]. The activations of the Fe-SMA bars were done one-by-one due to the practical limitation of the electric power supplier used in the experiment. In this section, the possible influence of the activation sequence on the prestressing effect and the bending resistance of the RC beam reinforced by the Fe-SMA bars was analyzed by the FE simulations. The same model used in Section 4.1, i.e., the FE model for the RC beam with 5 Fe-SMA bars, was used. The Fe-SMA bars in the FE simulations in this section were sequentially activated in the same order as in the experiment [31]. The FE simulation results were compared with the results of the same model but with activation of all 5 Fe-SMA bars activated at once.

Figure 8 shows the normal $x$-axis stress distributions along each Fe-SMA bar after activation, calculated by the FE simulations. Figure 8a shows the normal stress distributions of the Fe-SMA bars in the case of sequentially activated Fe-SMA bars. The activations of the Fe-SMA bars were carried out in the order of SMA1, SMA2, . . . , SMA5 in the simulations, as schematically shown by an insert of the figure. The normal stress distributions of the Fe-SMA bars in the case of a single activation are shown for comparison in Figure 8b. It is worth noting that, during the activation, a thermal contraction of 0.002233 mm/mm along the $x$-direction was applied in each Fe-SMA bar. The corresponding prestress when the Fe-SMA bar is strictly restrained along the $x$-direction is ~335 MPa. As shown in Figure 8a, it can be seen that the stress produced by the Fe-SMA bars are about 320–323 MPa in the case of the sequential activation. Therefore, the prestress obtained by the sequential activation of the Fe-SMA bars is about 10% less than the maximum prestress that can be obtained by the Fe-SMA bars with fully constrained condition. In this case, the $x$-axis stress along the Fe-SMA bars is almost the same without pronounced variation along the $x$-direction, except near the end of the RC beam where the stress drops sharply to zero due to the absence of the constraining concrete material outside the RC beam. When the Fe-SMA bars were activated at once, the $x$-axis stress near the center of the RC beam was about 335 MPa, which corresponds very well to the maximum prestress that can be obtained by the activation of the Fe-SMA. The stress decreased moderately and linearly with increasing the distance from the center, and then sharply dropped to zero near the end of the RC beam, as shown in Figure 8b.

The results in Figure 8 indicate that the sequential activation of the Fe-SMA bars produces slightly less prestress than the activation of the Fe-SMA bars at once. Near the center of the RC beam, the Fe-SMA bars activated at once produced $x$-axis stress very close to the ideal value of 335 MPa, while the sequentially activated Fe-SMA bars produced stress approximately 10% less than the prestress obtainable by the fully constrained condition.

The moderate decrease in the prestress in Figure 8b with increasing the distance from the center of the RC beam in the Fe-SMA can be easily understood by the elastic shortening effect of the concrete. While the concrete material is elastically shortened during the activation of the Fe-SMA bars, the bars are not under fully fixed conditions but under elastic boundary conditions. Therefore, the Fe-SMA bars are shortened especially near the end of the RC beam due to the upward bending and $x$-direction shortening of the RC beam, which can reduce the prestress produced by the Fe-SMA bars in comparison to the ideal, fully constrained condition [37]. In the case of sequential activation, the elastic shortening of the concrete can occur in a more complex manner depending on the three-dimensional positions of each Fe-SMA bar and the order of the activation. For instance, the activation of SMA2 in Figure 8a will distort the concrete beam not only in the $y$-direction but also in the $z$-direction. The bending distortion due to the SMA2 in the $y$-direction will shorten the SMA1 and relax the prestress of the SMA1 while the bending in the $z$-direction will tighten and increase the prestress of the SMA1. Moreover, if the concrete is locally cracked

or crushed during these sequences, the degradation of the concrete properties will induce additional loss of the prestress in the Fe-SMA bars.

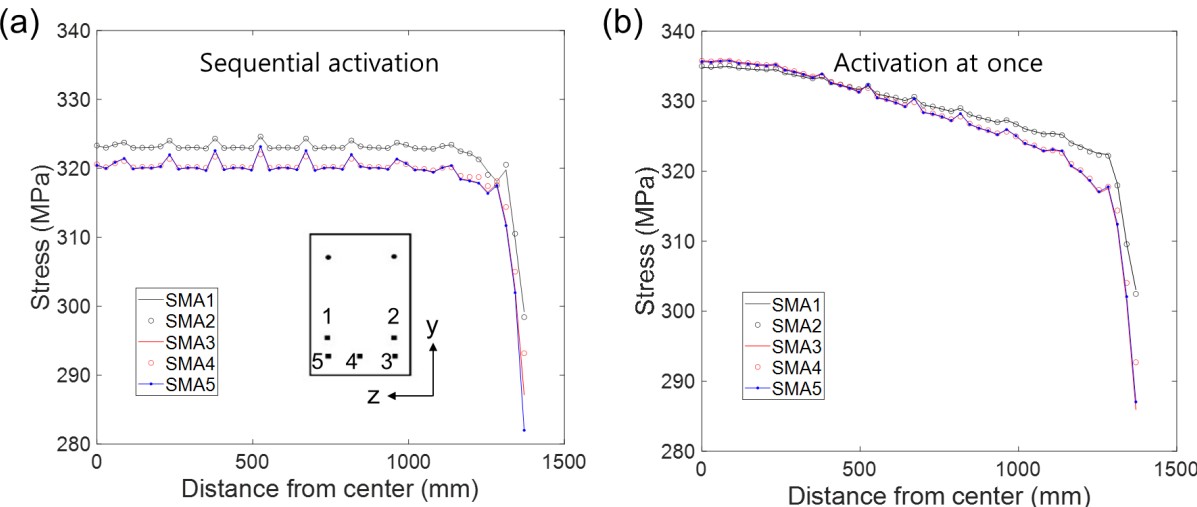

**Figure 8.** Normal *x*-axis stress distributions along Fe-SMA bars after activation, calculated by FE simulations. (**a**) With sequential activation of 5 Fe-SMA bars and (**b**) with 5 Fe-SMA bars activated at once.

Figure 9 shows the concrete cracking patterns due to the activation of the Fe-SMA bars predicted by the FE simulations, with sequential activation and activation at once. It evidently shows that the sequential activation of the Fe-SMA bars causes more damage in the concrete near the end of the RC beam than the activation at once. The activation of Fe-SMA bars induced damage of the concrete mainly near the bottom center of the RC beam and near the end of the RC beam (i.e., at regions A and B in Figure 9). The two FE models produced almost identical cracking patterns at region A, whereas pronouncedly severe shear-induced inclined cracks appeared in the sequential activation in comparison to the activation at once at region B. Tthe more severe cracking in this region in the sequential activation than the activation at once is probably due to the complex *y*- and *z*-direction distortions during the sequential activation of the Fe-SMA bars. The cracking patterns in this region influenced the loss of the prestress in the sequentially activated Fe-SMA bars due to the degradation of the concrete and consequently led to the lower *x*-axis stress in the Fe-SMA bars after the sequential activation in comparison to the activation at once.

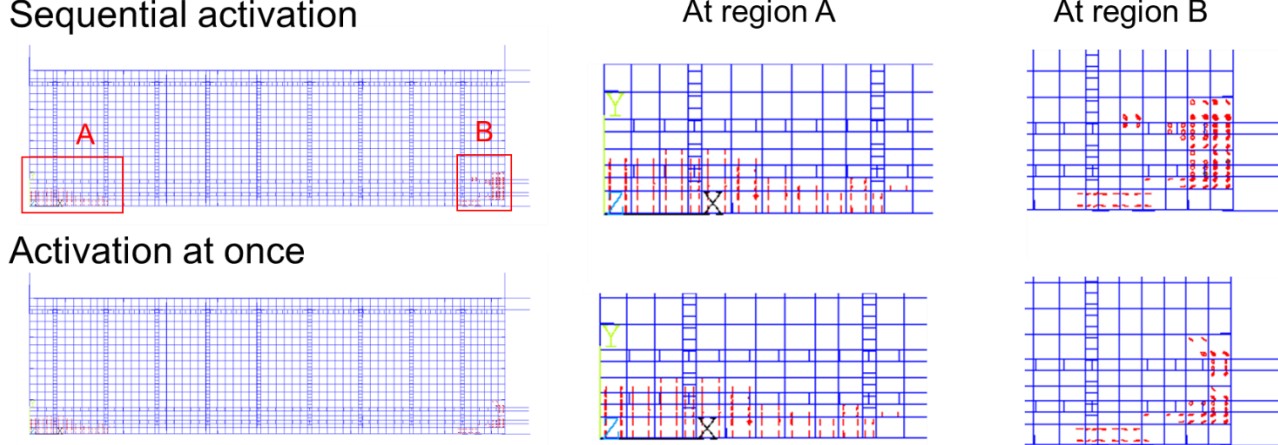

**Figure 9.** Concrete cracking patterns on *x-y* plane developed during activation of 5 Fe-SMA bars, with sequential activation and with activation at once.

Regardless of the difference in the crack patterns and the *x*-axis stress distributions of the Fe-SMA bars, there was no clear difference in the bending load-deflection responses in the two different FE models with the sequential activation and the activation at once, as shown in Figure 10. The load-deflection curve obtained by the FE model with the sequential activation followed nearly the same trend as that of the activation at once, indicating that the effect of Fe-SMA bar activation in sequence is negligible on the bending behavior of the RC beam prestressed by the Fe-SMAs.

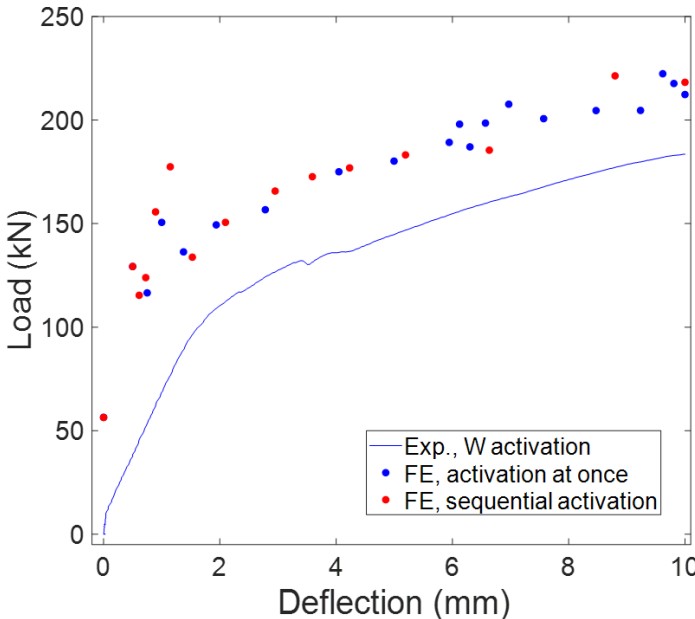

**Figure 10.** Comparison of experimental results and FE simulations for bending load-deflection responses of RC beams with 5 SMA bars, activated sequentially and at once.

### 4.3. Analysis of Bending Behavior of RC Beams with Different Number of SMA Bars

In this section, the bending behaviors of the RC beams prestressed with different numbers of the Fe-SMA bars were analyzed with the developed FE simulation models and the results are compared with the experiments. The RC beams considered in this section are the beams prestressed by 4, 3 and 2 Fe-SMA bars. The details of the RC beams considered in this section have been described in Section 2. The geometries and FE meshes of the FE simulation model used in Sections 4.1 and 4.2 are slightly modified according to the real geometries used in the experiment described in Section 2, and used for the FE simulations.

The experimentally determined bending load-deflection curves of the RC beams prestressed by different numbers of Fe-SMA bars are shown in Figure 11. In the figures, the bending load-deflection curves of the same RC beams but without Fe-SMA activation are also shown for comparison. It can be clearly seen that the activation of the Fe-SMA bars can effectively produce prestresses in the RC beams. The resistances to the bending deformations of the RC beams increased significantly with the activations of the Fe-SMA bars, regardless of the number of Fe-SMA bars used. The prestressed RC beams showed much higher initial crack loads than the same beams without the activation, as shown in Figure 11b.

Figure 12a shows the bending load-deflection curves predicted by the FE simulations for the RC beams with various numbers of activated Fe-SMA bars. The corresponding load-deflection curves obtained by the experiments are shown in Figure 12b for comparison. Although some oscillations occurred in the load-deflection curves obtained by the FE simulations due to the inhomogeneous and localized cracking in the discretized FE elements for the concrete, the overall tendencies of the FE simulation results are very similar to those observed in the experiments. In Figure 12a, the curves move upward without significant

change in slope with an increasing number of Fe-SMA bars, in the FE simulations. The experimentally obtained load-deflection curves showed the same tendency, as shown in Figure 12b. This means that the developed FE model is able to qualitatively represent the bending behavior of the RC beams prestressed by different numbers of the Fe-SMA bars. Therefore, the developed FE model can be used for an optimization study to select the best possible design parameters for prestressing the RC beam with the Fe-SMA bars. On the other hand, the FE simulations slightly overestimate the load response of the RC beams by 20–30 kN for every case. This is probably due to the imperfect bonding between the Fe-SMA bars and the concretes in the real experiments, which was not considered in the FE simulations of this study, as has been discussed in Section 4.1.

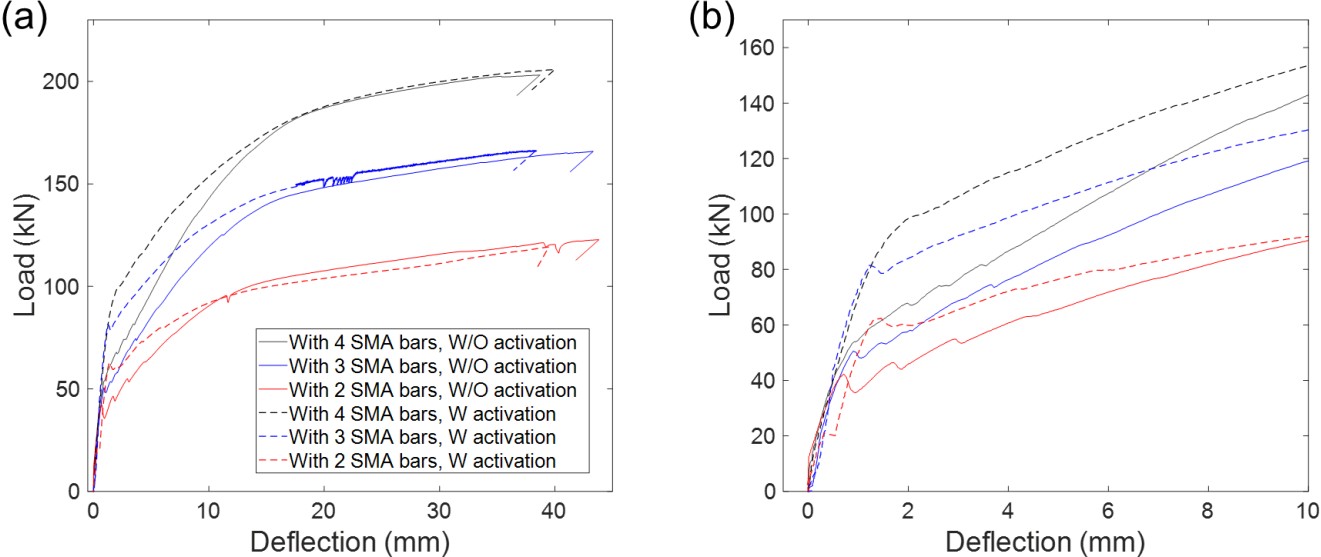

**Figure 11.** Experimental bending load-deflection curves of RC beams with various numbers of Fe-SMA bars, with and without Fe-SMA activation. (**a**) Overall load-deflection curves up to fracture and (**b**) enlarged view in deflection range of 0–10 mm.

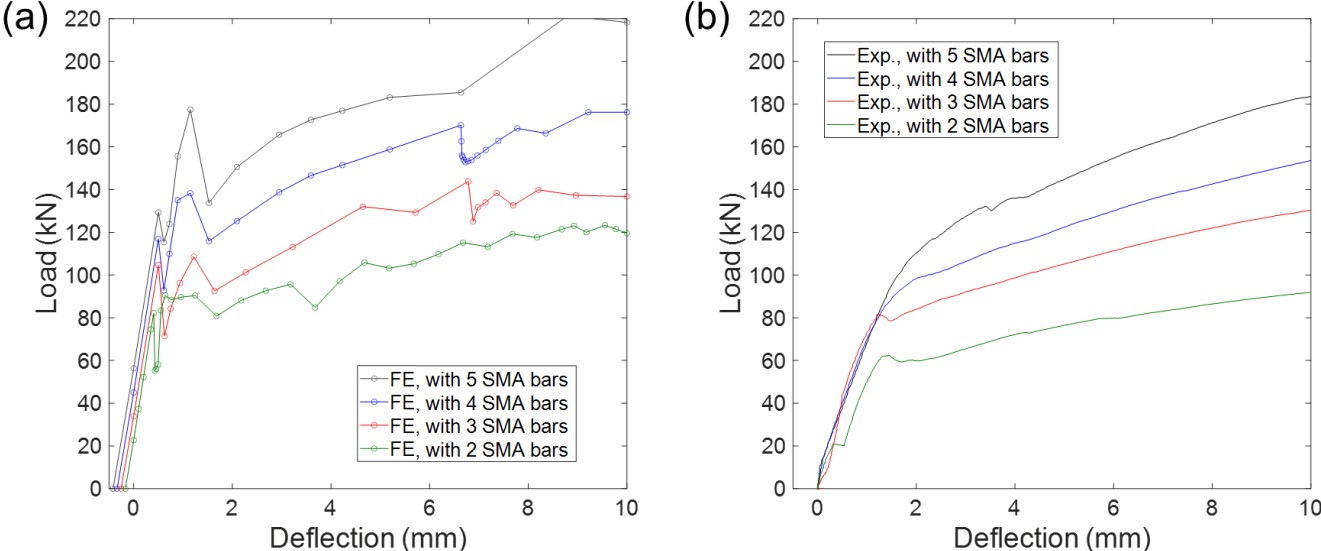

**Figure 12.** Bending load-deflection curves of RC beams with various numbers of Fe-SMA bars with activation, (**a**) predicted by FE simulations and (**b**) by experiments.

In Table 1, the RC beam cambers due to the activations of the Fe-SMA bars measured by the experiments [31] are compared with the predicted cambers by the FE simulations. In

the case of the RC beam with 5 Fe-SMA bars, it was found that the effect of the sequence of the Fe-SMA activations on the camber of the RC beam is found to be negligible as the FE models with the sequential activation and with the activation at once produced the same cambers of 0.423 mm. The camber predicted by the FE simulation matches the RC beam prestressed by 5 Fe-SMA bars very well, and the predicted values become slightly larger than those observed in the experiments with decreasing the number of the Fe-SMA bars. This again indicates that the FE simulation models used in this study slightly overestimate the prestress introduced by the activation of the Fe-SMA bars, due to the highly idealized assumptions of the perfectly bonded interfaces. However, the predicted RC beam cambers are in overall very good agreement with the experimental data with errors less than 0.052 mm.

**Table 1.** RC beam cambers due to activation of Fe-SMA measured by experiments [31] and calculated by FE simulations.

|  |  | Number of SMA Bars | | | |
|---|---|---|---|---|---|
|  |  | **5** | **4** | **3** | **2** |
| RC Beam camber (mm) | FE simulations | 0.423 | 0.340 | 0.258 | 0.171 |
|  | Experiments | 0.420 | 0.297 | 0.206 | 0.140 |

Figure 13 shows the normal *x*-axis stress distributions along each Fe-SMA bar after activations, calculated by the FE simulations for the RC beams prestressed by different numbers of Fe-SMA bars. It can be clearly seen that the prestress induced by the Fe-SMA bars tends to increase slightly with decreasing the number of Fe-SMA bars. Especially, in the case of the RC beam with 2 Fe-SMA bars, the stress distributions in the Fe-SMA bars are very close to the ideal value of 335 MPa (i.e., prestress that can be obtained by activating the Fe-SMA bar in the fully constrained condition) over nearly the entire length of the bars, as shown in Figure 13c. This confirms the idea that the loss of prestress in the studied RC beam systems is due to the combined effects of degradation of the concrete and the elastic shortening during the activation, as the number of Fe-SMA bars is reduced, the cracking as well as the elastic shortening of the concrete during the activation are both reduced.

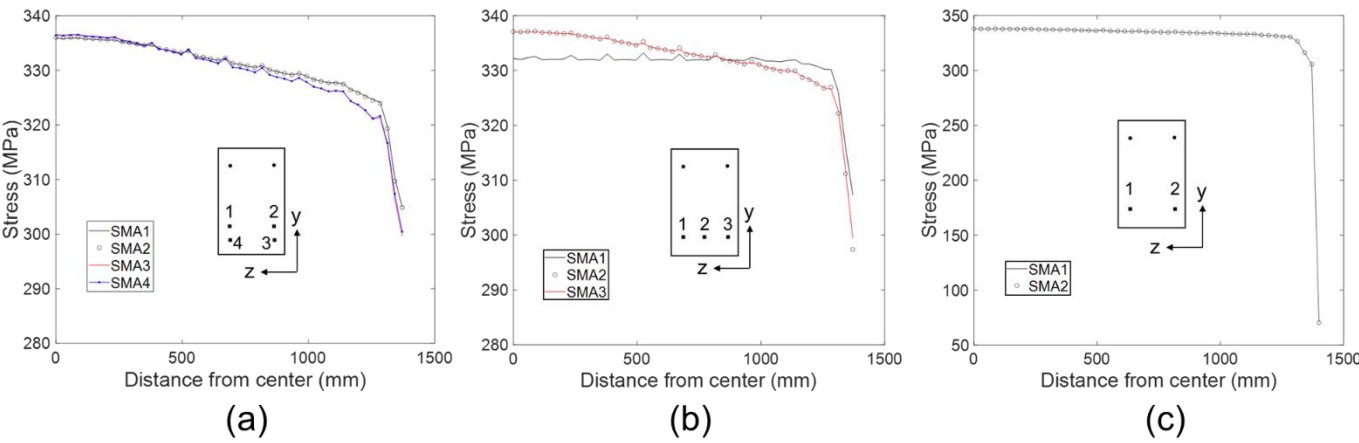

**Figure 13.** Normal *x*-axis stress distributions along Fe-SMA bars after activation, calculated by FE simulations with various numbers of Fe-SMA bars. (**a**) With 4 Fe-SMA bars, (**b**) with 3 Fe-SMA bars and (**c**) with 2 Fe-SMA bars.

Evolutions of equivalent stress fields of the concretes with bending deformations, predicted by the FE simulations of the RC beams prestressed with different numbers of the Fe-SMA bars, are shown in Figure 14. It can be seen that the overall stress fields in the RC beams with different numbers of the Fe-SMA bars are very similar, indicating that the stress in the concrete during bending is controlled mainly by the applied bending strain. When the applied bending deflection was higher than 2 mm, a high compressive stress region appears at the upper part of the RC beam near the center due to the bending of the RC beam. At a relatively large bending deflection of 10 mm, heterogeneously distributed localized high stress regions appear near the center of the RC beams, which are indicative of local bending due to the cracking of the concrete. These cracked regions are widely formed in the RC beam prestressed by 4 Fe-SMA bars, and it becomes narrower with decreasing the number of activated Fe-SMA bars. This means that the smaller number of activated Fe-SMA bars causes more concentrated distribution of serious cracking and crushing of the concrete, due to the less prestressing effect.

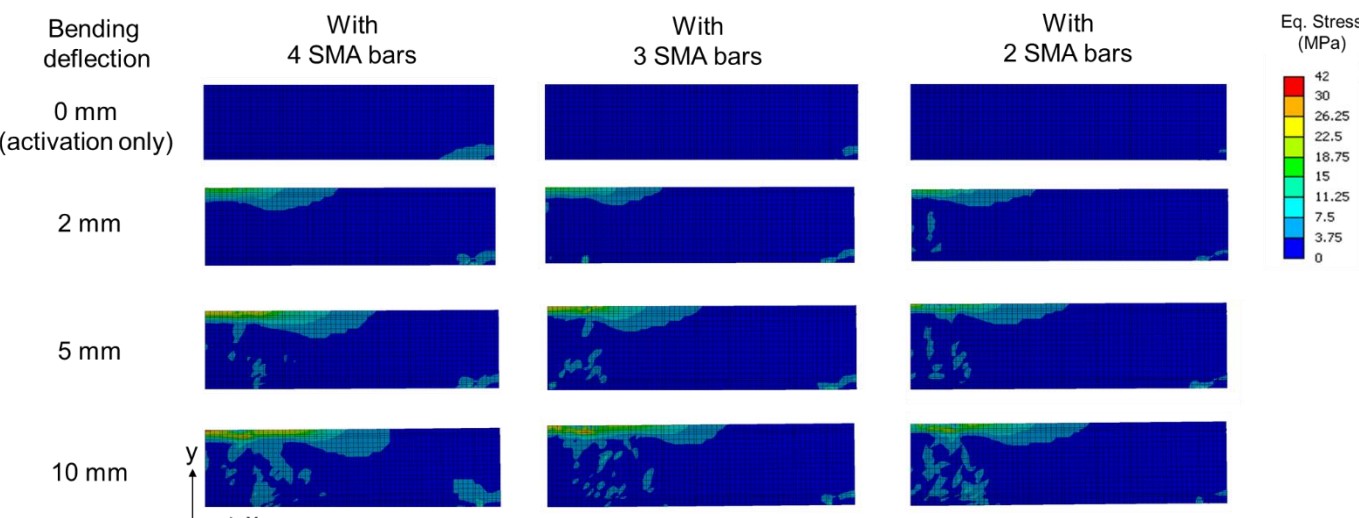

**Figure 14.** Evolutions of equivalent stress fields of concrete with bending predicted by FE simulations of RC beam with activated 4, 3 and 2 Fe-SMA bars.

Figure 15 shows the concrete cracking and crashing patterns predicted by the FE simulations, for the RC beams prestressed with different numbers of the Fe-SMA bars. The regions where the concrete is seriously damaged becomes narrower and narrower with a decreasing number of Fe-SMA bars at the same bending deflection, indicating once again that increasing the number of activated Fe-SMA bars can result in more distributed cracking pattern due to the prestressing effect. Thus, this effect together with the load transfer to the stiff Fe-SMA bars resulted in the increasing bending strength of the RC beam with increasing the number of the activated Fe-SMA bars, as observed in both the FE simulations and experiments in Figure 12. The experimentally observed cracking patterns shown in Figure 16 match very well with the FE simulation results, where it can be clearly seen that the cracks developed more severely in more concentrated distribution when a lesser number of activated Fe-SMA bars were used.

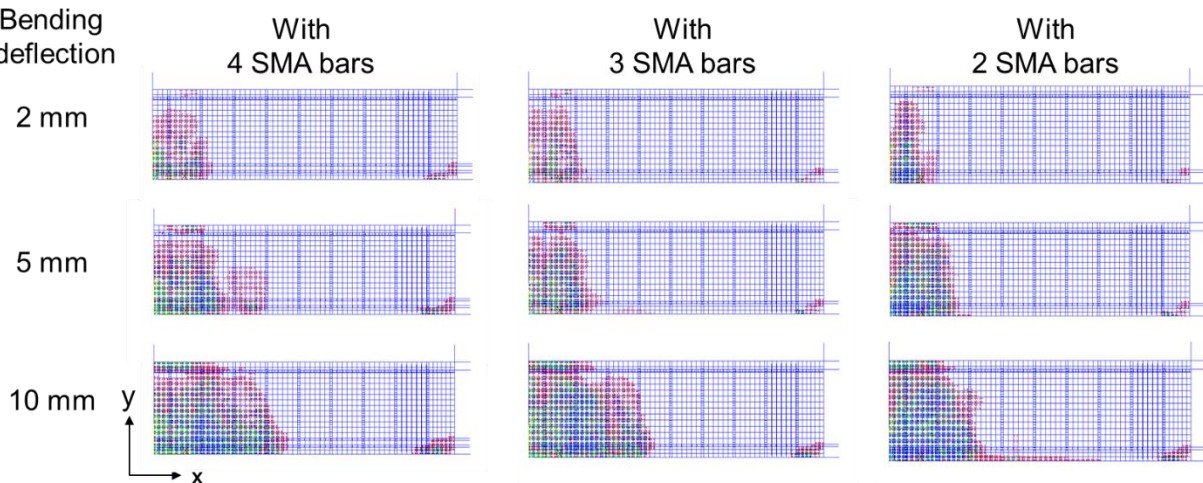

**Figure 15.** Concrete cracking and crushing patterns on *x-y* plane developed during RC beam bending, with activated 4, 3 and 2 Fe-SMA bars.

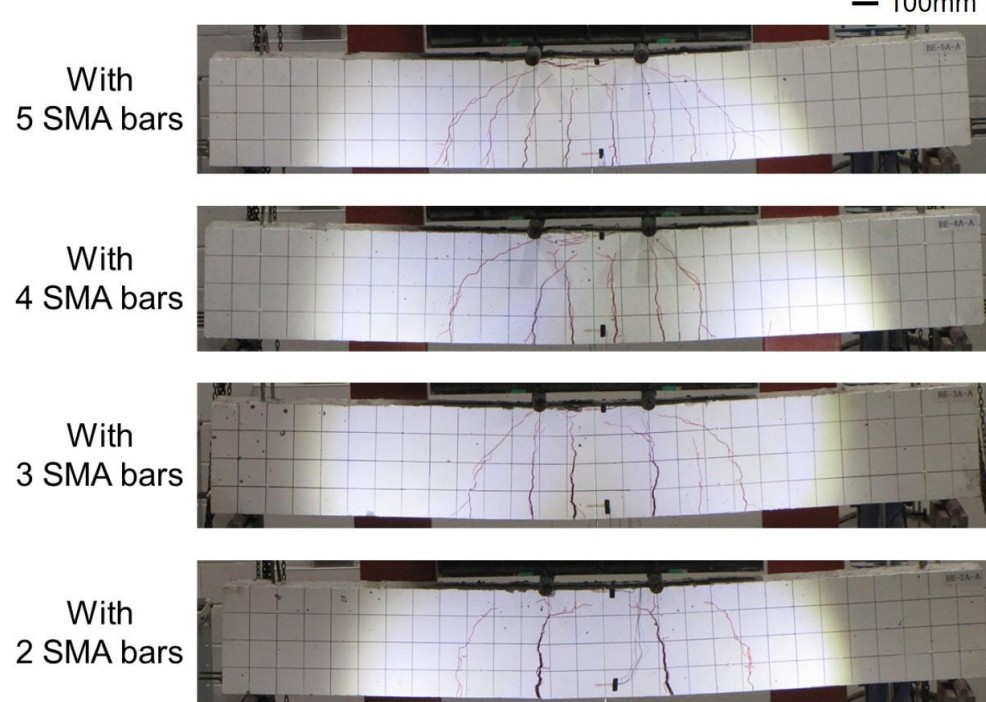

**Figure 16.** Experimentally observed crack patterns of RC beams after bending fracture with deflection of 40 mm, with various numbers of activated Fe-SMA bars of 5, 4, 3 and 2 [31].

## 5. Conclusions

In this study, the prestressing effect by the Fe-SMA bars of the RC beams was investigated both numerically and experimentally. The FE simulation model was developed to investigate the bending responses of the RC beams including nonlinear material properties such as concrete cracking and crushing as well as the plastic deformation of the Fe-SMA bars. By interpreting the numerical results and comparing them with the experimental observations for the RC beams prestressed by the Fe-SMAs in different prestressing scenarios, the following conclusions are drawn:

- A FE simulation model was developed to investigate the prestressing effect by the Fe-SMA bar on the RC beam bending deformation. The developed FE model took account of concrete cracking and crushing behavior, the prestressing effect by thermal activation of Fe-SMA bars, and the non-linear response of Fe-SMA bars.

- The FE model developed in this study was able to capture the bending behavior of the RC beam prestressed with the Fe-SMA bars. The prestressing effect was clearly represented in the bending load-deflection curve predicted by the FE simulations. However, the FE model slightly overestimated the prestressing effect by the Fe-SMA bars. The bending load predicted by the FE simulations was 20–30 kN higher than that obtained in the real experiments. This was thought to be due to the highly idealized perfect bonding assumption between the Fe-SMA bars and the concrete used in the FE simulations.

- The prestress induced by the activation of the Fe-SMA bars was influenced only slightly by the sequence of the activations of multiple Fe-SMA bars. When interpreting the FE simulation results, this was due to the complex bending distortions of the RC beam occurring during the sequential activation of the Fe-SMA bars. However, the bending response of the RC beam was not significantly influenced by the sequence of the activation. Therefore, the activation sequence can be neglected in the FE simulations for calculating the bending deformation responses.

- FE simulations of the RC beams prestressed by different numbers of the Fe-SMA bars qualitatively reproduced the experimentally observed bending deformation behaviors very well. The cracking behaviors of the concrete beams strengthened by prestressed Fe-SMA bars during bending are also well represented by the developed FE models. Although the FE model developed in this study slightly overestimates the prestressing effect, the developed FE model can be used for an optimization study to select the best possible design parameters for prestressing the RC beam with the Fe-SMA bars.

**Author Contributions:** Conceptualization, Y.-M.Y. and W.L.; methodology, Y.-M.Y.; software, W.L.; validation, Y.-M.Y. and K.-N.H.; investigation, Y.-M.Y.; resources, K.-N.H.; writing—original draft preparation, Y.-M.Y.; writing—review and editing, W.L. and K.-N.H. All authors have read and agreed to the published version of the manuscript.

**Funding:** This work was supported by Pusan National University Research Grant, 2021.

**Institutional Review Board Statement:** Not applicable.

**Data Availability Statement:** The data presented in this study are available on request from the corresponding author, upon reasonable request.

**Conflicts of Interest:** The authors declare no conflict of interest.

## Abbreviations

| Abbreviations | | Symbols | |
| --- | --- | --- | --- |
| EB | External bonded | $E_c$ | Elastic modulus of the concrete |
| Fe-SMA | Fe-based shape memory alloy | $\varepsilon$ | Strain of the concrete |
| FE | Finite Element | $\varepsilon_0$ | Strain corresponding to compressive strength of the concrete |
| LVDT | Linear-variable displacement | $f_1$ | Compressive strength for a state of biaxial compression superimposed on hydrostatic stress state |
| Nitinol | Ni-Ti alloy | $f_2$ | Uniaxial compression superimposed on hydrostatic stress state |
| NSM | Near-surface mounted | $f_c$ | Compressive stress of the concrete |
| RC | Reinforced concrete | $f_{cb}$ | Biaxial compressive strength of concrete |
| SMA | Shape memory alloy | $f_t$ | Uniaxial tensile strength of concrete |
| SME | Shape memory effect | $f'_c$ | Uniaxial compressive strength of concrete |

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
