# Peer review of "Finite Element Analysis of Reinforced Concrete Beams Prestressed by Fe-Based Shape Memory Alloy Bars"

_applsci, doi:10.3390/app12073255_

Round 1

Reviewer 1 Report

Thank you for submitting your paper. The work done here draws attention to a significant subject FE modelling of concrete structures. I have found the paper to be interesting. However, several issues need to be addressed properly before the paper is being considered for publication. My comments including major and minor concerns are given below:

  1. Please consider reviewing the abstract and highlight the novelty, major findings, and conclusions. I suggest reorganizing the abstract, highlighting the novelties introduced. The abstract should contain answers to the following questions:
  2. What problem was studied and why is it important?
  3. What methods were used?
  4. What conclusions can be drawn from the results? (Please provide specific results and not generic ones).
  5. The abstract must be improved. It should be expanded. Please use numbers or % terms to clearly shows us the results in your experimental work.
  6. Please consider reporting on studies related to your work from mdpi journals.
  7. The introduction must be expanded, please consider improving the introduction, provide more in-depth critical review about past studies similar to your work, mention what they did and what were their main findings then highlight how does your current study brings new difference to the field.
  8. Please combine all small paragraphs into larger ones.
  9. Materials and Methods section lacks any photos or figures for material setup, test equipment used and other relevant images which can better show what was done in this work, after all this is an experimental study and graphical details are important to support the information provided in the text.
  10. It is better to separate the FE section from the experimental section. Make section 3 Materials and methods, and section 4 Finite element modelling
  11. Line 108, I am confused, so the authors did not do any experimental work, you used data from a previous study?
  12. Line 139 “of the FE models are set to the same to the previous work” please check this sentence it does not read well.
  13. Please add a table at the end of the manuscript which contains all abbreviations, Greek letters and symbols used in this study.
  14. Line 183 “For the Fe-SMA bars, experimentally determined elastic modulus of 150 GPa” why are you measuring E, why not use the one from the experimental work from previous study, this is not good practice as the results this way might be incorrect. What was the E from the previous study?
  15. Lines 176-194 combine into one larger paragraph.
  16. Figure 4 there is a significant discrepancy between the FE and Exp results, why?
  17. The authors should mention some of the limitations in the FE model which might affect the accuracy of the predictions.
  18. Line 333 “slightly less prestress than the activation” by how much? 1% 10% or more, please be specific. Also how about past studies, did they report similar observations in their FE models. Could there be other underlying factors which affect the results? Please discuss and support with references.
  19. Enlarge figure 8 so we could better see the crack patterns. Use landscape layout.
  20. Please add scale bar to figure 15.
  21. Some of the results are merely described and is limited to comparing the experimental observation and describing results. The authors are encouraged to include a more detailed results and discussion section and critically discuss the observations from this investigation with existing literature.
  22. Conclusion can be expanded or perhaps consider using bullet points (1-2 bullet points) from each of the subsections.

Author Response

The authors thank to the editor and the reviewers for their careful read and thoughtful comments on our paper, which helped us very much in improving the manuscript. We have carefully taken all the comments into consideration in preparing our new manuscript. In the new manuscript, revised texts are marked with the ‘track changes’ function of the Microsoft word. The comments are given in the following sections in bold face, while the authors' responses are given in normal font.

Thank you for submitting your paper. The work done here draws attention to a significant subject FE modelling of concrete structures. I have found the paper to be interesting. However, several issues need to be addressed properly before the paper is being considered for publication. My comments including major and minor concerns are given below:

  1. Please consider reviewing the abstract and highlight the novelty, major findings, and conclusions. I suggest reorganizing the abstract, highlighting the novelties introduced. The abstract should contain answers to the following questions:

What problem was studied and why is it important?

What methods were used?

What conclusions can be drawn from the results? (Please provide specific results and not generic ones).

The abstract must be improved. It should be expanded. Please use numbers or % terms to clearly shows us the results in your experimental work.

Answer: We have revised and expanded our abstract to address the reviewer’s concerns. Thank you very much for the comment. The revised abstract is as follows:

“Prestressing of concrete structures using Fe-based shape memory alloys have been investigated extensively by experiments in the last decade. However, detailed investigations on the stress produced by the Fe-based shape memory alloys and its influence on concrete damage during deformation of concrete structure has not been investigated yet. In this study, pre-stressing effect by Fe-based shape memory alloy bars on bending behavior of reinforced concrete beam is investigated both numerically and experimentally. Finite element simulation model is developed to investigated the bending responses of the beams including nonlinear material properties such as concrete cracking and crushing as well as the plastic deformation of the Fe-based shape memory alloy. The model is able to capture the bending behavior of the beam prestressed with the Fe-based shape memory alloy bars. Based on the numerical and experimental results, the prestressing effect by the shape memory alloy bars are investigated in detail. Although the developed model slightly overestimated the experimentally obtained bending load-deflection curves of the concrete beams, it is shown that the developed model can be used for an optimization study to select the best possible design parameters for pre-stressing the concrete beam with the Fe-based shape memory alloy bars. A possible reason for the overestimation is the idealized perfect bonding assumption between Fe-SMA and concrete used in the model, while slip at the interface occurred in the experiments.”

  1. Please consider reporting on studies related to your work from mdpi journals.

The introduction must be expanded, please consider improving the introduction, provide more in-depth critical review about past studies similar to your work, mention what they did and what were their main findings then highlight how does your current study brings new difference to the field.

Please combine all small paragraphs into larger ones.

Answer: Thank you very much for the nice comment. In regard to this comment about the introduction, we have mentioned all of our previous works, which are related to the use of Fe-based shape memory alloys, published in the MDPI journals in the introduction and in the main content of the paper. The corresponding references are Ref 25, 26 and 27. Additionally, we have added five more references in the revised manuscript to address better the previous works done in this field, including one additional work published previously in the MDPI journal. We have also merged some small paragraphs into larger ones. As it does not directly appear in the revised text due to the ‘track changes’ options, the revised introduction has only three large paragraphs (The changes can be seen directly when accepting all the changes in the word file). We hope that the revised manuscript can meet the reviewer’s requirements.

  1. Materials and Methods section lacks any photos or figures for material setup, test equipment used and other relevant images which can better show what was done in this work, after all this is an experimental study and graphical details are important to support the information provided in the text.

Answer: Thank you very much for the nice comment. In accordance to this comment, we have added one additional figure in the revised paper with the explanations. The below is added:

Figure 2. Photos of (a) Fe-SMA bars used for the experiments, (b) Fe-SMA and stirrup assembly, (c) molding for concrete casting and (d) concrete casting process. -please find the details in the attached file 

  1. It is better to separate the FE section from the experimental section. Make section 3 Materials and methods, and section 4 Finite element modelling

Answer: Done. Thank you very much for this nice comment.

  1. Line 108, I am confused, so the authors did not do any experimental work, you used data from a previous study?

Answer: We have used the experimental data obtained from the same specimens and tests described in Ref.[27], which is indeed the authors’ previous work. In the revised paper, we have clearly explained about this point. The following statements are added:

“In the authors’ previous work, 10 specimens were constructed to evaluate the bending behavior of concrete beams reinforced with Fe-SMA bars [27]. The experimental data obtained from the same specimens were used in this study. As shown in Fig.1, the width, effective depth and height of the specimens were 250 mm, 350 mm and 400 mm, respectively. The total length and net-span of the specimens were 2,800 mm and 2,600 mm, respectively. As described in [27], the previous experimental work considered Fe-SMA activation and amount of Fe-SMA bar (200 mm2, 300 mm2, 400 mm2, and 500 mm2) as experimental variables.”

  1. Line 139 “of the FE models are set to the same to the previous work” please check this sentence it does not read well.

Answer: Thank you very much for pointing out this point. The mentioned sentence has been revised as follows:

“Thus, the geometries of the FE models are designed to be the same to the geometries of the specimens used in the previous work [27]”

  1. Please add a table at the end of the manuscript which contains all abbreviations, Greek letters and symbols used in this study.

Answer: Thank you for the nice comment. Following table is added in the revised manuscript, at the end of the paper:

List of Symbols and Abbreviations

Abbraviations

Symbols

EB

External bonded

Ec

Elastic modulus of the concrete

Fe-SMA

Fe-based shape memory alloy

ε

Strain of the concrete

FE

Finite Element

ε0

Strain corresponding to compressive strength of the concrete

LVDT

Linear-variable displacement

f1

Compressive strength for a state of biaxial compression superimposed on hydrostatic stress state

Nitinol

Ni-Ti alloy

f2

Uniaxial compression superimposed on hydrostatic stress state

NSM

Near-surface mounted

fc

Compressive stress of the concrete

RC

Reinforced concrete

fcb

Biaxial compressive strength of concrete

SMA

Shape memory alloy

ft

Uniaxial tensile strength of concrete

SME

Shape memory effect

fc

Uniaxial compressive strength of concrete

  1. Line 183 “For the Fe-SMA bars, experimentally determined elastic modulus of 150 GPa” why are you measuring E, why not use the one from the experimental work from previous study, this is not good practice as the results this way might be incorrect. What was the E from the previous study?

Answer: We have indeed used the experimentally determined elastic modulus. This was approximately 150 GPa for the Fe-SMA bars, as described in ref.[27]. We are sorry for the confusion. To clarify this point, we have cited our previous work in the revised paper, as follows:

“For the Fe-SMA bars, experimentally determined elastic modulus of 150 GPa [27] with Poisson’s ratio of 0.33 was used”

  1. Lines 176-194 combine into one larger paragraph.

Answer: Done. Thank you very much!

  1. Figure 4 there is a significant discrepancy between the FE and Exp results, why?

The authors should mention some of the limitations in the FE model which might affect the accuracy of the predictions.

Answer: We have already explained the possible reason for the discrepancy between the FE model and the experiment in the text. The possible reason can be that highly idealized perfect bonding between the Fe-SMA bars and the concrete was assumed in the FE simulations, while there can be some slip at the interface during the bending deformation. The corresponding statements in the text are as follows:

“One possible explanation for the overestimated prestress of the Fe-SMA is the imperfect bonding between the Fe-SMA bars and the concrete in the experiments. In the experiments, there was no anchoring between the concrete and the Fe-SMA bars and the Fe-SMA bars had nearly flat surfaces [27]. If there was debonding or slip of the interfaces between the Fe-SMA bars and the concrete in the experiments, this could make the present FE simulation to have overestimated prestress state while activating the Fe-SMA bars where perfect bonding between the bars and the concrete was assumed. To take into account the realistic bonding behavior between the Fe-SMA bars and the concrete, sophisticated FE simulations with consideration of bonding-debonding [32] or with specialized interface elements for the consideration of slip [31] are needed, which is beyond the scope of current FE simulations.”

To address the above point more clearly in the revised text, we have made a separated paragraph for these important explanations.

  1. Line 333 “slightly less prestress than the activation” by how much? 1% 10% or more, please be specific. Also how about past studies, did they report similar observations in their FE models. Could there be other underlying factors which affect the results? Please discuss and support with references.

Some of the results are merely described and is limited to comparing the experimental observation and describing results. The authors are encouraged to include a more detailed results and discussion section and critically discuss the observations from this investigation with existing literature.

Answer: With respect to the reviewer, we would like to clarify that the issues that had been raised by the reviewer at this point are all already addressed in the paper. We guess that the reviewer could not find them during the reviewing process. Prestresses according to the two different prestressing scenarios are quantitatively compared in just three lines below the line pointed out by the reviewer. The difference was ~10 %. In the next paragraph, the possible factors which affect the results are discussed in detail with references. Please find the following statements in the revised paper:

“The moderate decrease in the prestress in Fig.7(b) with increasing the distance from the center of the RC beam in the Fe-SMA can be easily understood by the elastic shortening effect of the concrete. While the concrete material is elastically shortened during the activation of the Fe-SMA bars, the bars are not under fully fixed conditions but under elastic boundary conditions. Therefore the Fe-SMA bars are shortened especially near the end of the RC beam due to the upward bending and x-direction shortening of the RC beam, which can reduce the prestress produced by the Fe-SMA bars in comparison to the ideal, fully constrained condition [33]. In the case of sequential activation, the elastic shortening of the concrete can occur in more complex manner depending on the three-dimensional positions of each Fe-SMA bar and the order of the activation. For instance, the activation of SMA2 in Fig.7(a) will distort the concrete beam not only in y-direction but also in z-direction. The bending distortion due to the SMA2 in the y-direction will shorten the SMA1 and relax the prestress of the SMA1 while the bending in the z-direction will tighten and increase the prestress of the SMA1. Moreover, if the concrete is locally cracked or crushed during these sequences, the degradation of the concrete properties will induce additional loss of the prestress in the Fe-SMA bars.”

  1. Enlarge figure 8 so we could better see the crack patterns. Use landscape layout.

Answer: Done. Thank you very much!

  1. Please add scale bar to figure 15.

Answer: Done. Thank you very much!

  1. Conclusion can be expanded or perhaps consider using bullet points (1-2 bullet points) from each of the subsections.

Answer: Thank you very much for the nice comment. We have improved and expanded our conclusions in the revised paper.

Reviewer 2 Report

In this manuscript, a numerical finite element analysis was conducted to investigate the pre-stressing effect of SMA bars on the flexural behaviour of RC beams. Although the advantage of shape memory alloys is the zero residual strains, in this study just the monotonic loadings were investigated based on a previous experimental program. Some questions and suggestions are provided below to enrich the scientific level of the presented manuscript before its publication:

  • As the manuscript conducted just numerical FEM based on previous experimental program, the first sentence of the abstract should be modified: “Prestressing effect by Fe-based shape memory alloy bars on bending behavior of reinforced concrete beam is investigated both numerically and experimentally.”
  • There are various new research works that studied the effect of SMA in structural behaviour that could be added to the introduction. Here are some suggestions to review:
    • Hosseini, A., Michels, J., Izadi, M., & Ghafoori, E. (2019). A comparative study between Fe-SMA and CFRP reinforcements for prestressed strengthening of metallic structures. Construction and Building Materials, 226, 976-992.
    • Farhangi, V., Jahangir, H., Eidgahee, D. R., Karimipour, A., Javan, S. A. N., Hasani, H., ... & Karakouzian, M. (2021). Behaviour Investigation of SMA-Equipped Bar Hysteretic Dampers Using Machine Learning Techniques. Applied Sciences, 11(21), 10057.

Suhail, R., Amato, G., Broderick, B., Grimes, M., & McCrum, D. (2021). Efficacy of prestressed SMA diagonal loops in seismic retrofitting of non-seismically detailed RC beam-column joints. Engineering Structures, 245, 112937.

  • Jahangir, H., & Bagheri, M. (2020). Evaluation of seismic response of concrete structures reinforced by shape memory alloys. International Journal of Engineering, 33(3), 410-418.
  • Choi, E., Ostadrahimi, A., Kim, W. J., & Seo, J. (2021). Prestressing effect of embedded Fe-based SMA wire on the flexural behavior of mortar beams. Engineering Structures, 227, 111472.
  • As the respectful authors know, the most important advantage of SMAs is their near-zero residual strain in cyclic loading. It would enrich the scientific level of the manuscript if this feature of SMA could be presented by applying cyclic loadings on the RC beams in numerical FEM models. This behaviour can be shown in Figure 3b.
  • The presented results in Figures 4 and 9 are not so consistent. What is the reason?
  • As this study just conducted numerical modeling, Figure 15 should be referenced to [27].

Author Response

The authors thank to the editor and the reviewers for their careful read and thoughtful comments on our paper, which helped us very much in improving the manuscript. We have carefully taken all the comments into consideration in preparing our new manuscript. In the new manuscript, revised texts are marked with the ‘track changes’ function of the Microsoft word. The comments are given in the following sections in bold face, while the authors' responses are given in normal font.

In this manuscript, a numerical finite element analysis was conducted to investigate the pre-stressing effect of SMA bars on the flexural behaviour of RC beams. Although the advantage of shape memory alloys is the zero residual strains, in this study just the monotonic loadings were investigated based on a previous experimental program. Some questions and suggestions are provided below to enrich the scientific level of the presented manuscript before its publication: 

  • As the manuscript conducted just numerical FEM based on previous experimental program, the first sentence of the abstract should be modified: “Prestressing effect by Fe-based shape memory alloy bars on bending behavior of reinforced concrete beam is investigated both numerically and experimentally.”

Answer: Thank you very much for the nice comment. In the revised paper, the mentioned sentence has been modified as follows:

“In this study, prestressing effect by Fe-based shape memory alloy bars on bending behavior of reinforced concrete beam is investigated numerically.”

As we have added some additional sentences in the first part of the abstract according to the reviewer #1’s opinion, the above sentence is the third sentence in the revised manuscript.

  • There are various new research works that studied the effect of SMA in structural behaviour that could be added to the introduction. Here are some suggestions to review:

Hosseini, A., Michels, J., Izadi, M., & Ghafoori, E. (2019). A comparative study between Fe-SMA and CFRP reinforcements for prestressed strengthening of metallic structures. Construction and Building Materials, 226, 976-992.

Farhangi, V., Jahangir, H., Eidgahee, D. R., Karimipour, A., Javan, S. A. N., Hasani, H., ... & Karakouzian, M. (2021). Behaviour Investigation of SMA-Equipped Bar Hysteretic Dampers Using Machine Learning Techniques. Applied Sciences, 11(21), 10057.

Suhail, R., Amato, G., Broderick, B., Grimes, M., & McCrum, D. (2021). Efficacy of prestressed SMA diagonal loops in seismic retrofitting of non-seismically detailed RC beam-column joints. Engineering Structures, 245, 112937.

Jahangir, H., & Bagheri, M. (2020). Evaluation of seismic response of concrete structures reinforced by shape memory alloys. International Journal of Engineering, 33(3), 410-418.

Choi, E., Ostadrahimi, A., Kim, W. J., & Seo, J. (2021). Prestressing effect of embedded Fe-based SMA wire on the flexural behavior of mortar beams. Engineering Structures, 227, 111472.

Answer: We appreciate the reviewer’s suggestions. We have cited all the references that are suggested by the reviewer in the introduction and added additional explanations in the revised paper.

  • As the respectful authors know, the most important advantage of SMAs is their near-zero residual strain in cyclic loading. It would enrich the scientific level of the manuscript if this feature of SMA could be presented by applying cyclic loadings on the RC beams in numerical FEM models. This behaviour can be shown in Figure 3b.

Answer: With respect to the reviewer’s comment, we would like to clarify that our Fe-based SMA is not superelastic SMAs. There are two types of SMA, i.e. shape memory and superelastic SMAs. Among them, superelastic SMAs always go back to their original shape while the applied load is removed, thereby showing near-zero residual strain in cyclic loading. By the way, normal shape memory alloys go back to their original shapes only when they are heated. We have used the normal shape memory alloy which does go only back to its original shape while heating (Indeed we were using this special behavior to activate the prestress by heating of the SMA). Therefore if we present the unloading curve in the Fig.4(b), it would be just a straight line.  

  • The presented results in Figures 4 and 9 are not so consistent. What is the reason?

Answer: The results presented in former Fig.4 compares the bending deflection curves between the concrete beams with and without Fe-SMA activation (i.e. prestressing). In former Fig.9, the Fe-SMA bars are all activated, but in one model they are activated at once while in the other they are activated sequentially. The results show that the bending deflection of the concrete beam is influenced significantly with the activation of the Fe-SMA bars, but not significantly with the sequence of the activation processes.   

  • As this study just conducted numerical modeling, Figure 15 should be referenced to [27].

Answer: Thank you very much for the nice comment. We have added the reference in the figure caption, as suggested.

Round 2

Reviewer 1 Report

All questions answered and paper can be accepted

Reviewer 2 Report

All the suggestions were applied and all the questions were responded in the revised version of the manuscript and can be accepted in the current version.